# GraphChef: Learning the Recipe of Your Dataset

Peter Müller [1]   Lukas Faber [1]   Karolis Martinkus [1]   Roger Wattenhofer [1]

## Abstract

We propose a new graph model, GraphChef, that enables us to understand graph datasets as a whole. Given a dataset, GraphChef returns a set of rules (a recipe) that describes each class in the dataset. Existing GNNs and explanation methods reason on individual graphs not on the entire dataset. GraphChef uses decision trees to build recipes that are understandable by humans. We show how to compute decision trees in the message passing framework in order to create GraphChef. We also present a new pruning method to produce small and easy to digest trees. In the experiments, we present and analyze GraphChef's recipes for Reddit-Binary, MUTAG, BA-2Motifs, BA-Shapes, Tree-Cycle, and Tree-Grid. We verify the correctness of the discovered recipes against the datasets' ground truth.

## 1. Introduction

Graphs abstractly represent complex relational data in a myriad of applications, and play a crucial role in, e.g., chemistry, engineering, social sciences, or transportation. Graph Neural Networks (GNNs) have been successfully applying machine learning techniques to many graph-based domains. So far GNNs are used as black-box models to classify individual graphs, as in "Is this protein (represented as a graph) an enzyme?" In contrast, in our work we do not only want to understand individual graphs, but *whole* datasets, as in "What makes a protein an enzyme?" To answer questions about whole datasets, we need a form of abstract articulation, a "recipe" that describes precisely which proteins are enzymes. Decision trees seem suitable to articulate such a recipe because they can produce human-readable explanations. In this paper, we propose a novel architecture called *GraphChef* that combines the strengths of decision trees with graph neural networks. As a motivating example, we

discuss the GraphChef recipe for Enzymes in Figure 1.

Enzymes are easy to understand without intermediate layers. We show in other experiments that GraphChef works equally well for various graph benchmarks. In summary, our contributions are as follows:

- While traditional GNNs are based on synchronous message passing (Loukas, 2020), we propose a new layer that is inspired by a simplified distributed computing model known as the stone age model (Emek & Wattenhofer, 2013). In this model, nodes use a small categorical space for states and messages. The stone age model is simple and as such suitable for interpretation while retaining a high theoretical expressiveness. We call our new layer *dish* (DIfferentiable Stone-"H").

- We distill the multi-layer perceptrons in all dish layers to decision trees. We call the resulting model GraphChef. GraphChef abstractly expresses the reasoning behind a graph classification task (the classification recipe) with a series of decision trees.

- We propose a way to collectively prune the decision trees in GraphChef. Pruning may affect accuracy, but also gives simpler explanations. GraphChef hence allows for a trade-off between accuracy and simplicity.

- We show that we can also use the decision trees to compute node-level importance scores similar to orthodox GNN explanation methods.

- We test our proposed architecture on established GNN explanation benchmarks and real-world graph datasets. We show that our recipes are competitive in classification accuracy with traditional GNNs. We further validate that the proposed pruning methods considerably reduce tree sizes. Last, we demonstrate how to read GraphChef's recipes to find interesting insights in real-world datasets or flaws in existing explanation benchmarks.

- We provide a user interface for GraphChef.[1] This tool allows for the interactive exploration of the GraphChef recipes on the datasets examined in this paper. We provide a manual for the interface in Appendix C.

---

[1]ETH Zurich, Switerland. Correspondence to: Lukas Faber <lfaber@ethz.ch>.

*Workshop on Interpretable ML in Healthcare at International Conference on Machine Learning (ICML)*, Honolulu, Hawaii, USA. 2023.

---

[1]https://interpretable-gnn.netlify.app/

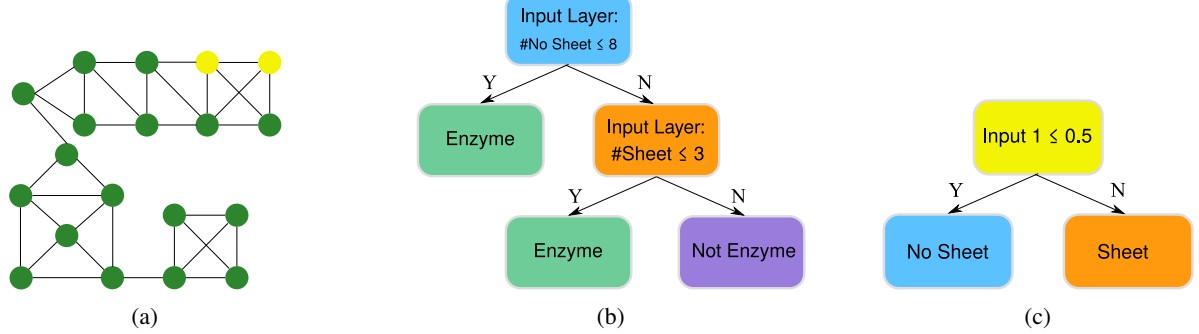

Figure 1: (a) An example graph from the PROTEINS dataset. In each graph of the dataset, nodes are secondary structural elements of amino acids. Each node has one of three types: helix (input 0), sheet (input 1), or turn (input 2). Figure (a) shows an example graph consisting of mostly helices (green) and two sheets (yellow). The output layer (b) learns to decide whether a protein is an enzyme based on how many sheet respectively "no sheet" nodes the graph has in the input layer. Whether nodes are helices or turns does not seem to matter, and consequently the input layer (c) learns to only distinguish nodes as either sheet and no sheet. Both trees (b,c) together then imply: *A protein is an enzyme if it has at most* 8 *nodes that are not sheets, or if it has at most* 3 *sheets*. This GraphChef recipe is consistent with previous human analysis (Errica et al., 2020).

## 2. Related Work

### 2.1. Explanation methods for GNNs

Given a graph classification problem, GraphChef will compute a series of decision trees (a recipe) that together explain the classification. As such, GraphChef is a distant relative of other explanation methods for GNNs. We can group existing explanation methods into roughly five types:

**Gradient based.** Baldassarre & Azizpour (2019) and Pope et al. (2019) show that it is possible to adopt gradient-based methods known from computer vision, for example Grad-CAM(Selvaraju et al., 2017). Gradients can be computed on node features and edges (Schlichtkrull et al., 2021).

**Mutual-information based.** Ying et al. (2019) measure the importance of edges and node features. Edges are masked with continuous values. Instead of gradients, the authors use mutual information between the inputs and the prediction to quantify the importance. Luo et al. (2020) follow a similar idea but emphasize finding structures that explain multiple instances at the same time.

**Counterfactual.** Counterfactual approaches measure the importance of nodes or edges by how much removing them changes the classifier prediction. They are the extension of the occlusion idea (Zeiler & Fergus, 2014) to graph neural networks. Lucic et al. (2021) for example identifies few edge deletions that change model predictions. Bajaj et al. (2021) propose a hybrid with an example-based explanation. They compute decision boundaries over multiple instances to find optimized counterfactual explanations.

**Subgraph based.** Yuan et al. (2021) consider each subgraph as possible explanation. To score a subgraph, they use Shapley values (Shapley, 1953) and Monte Carlo tree search for guiding the search. Duval & Malliaros (2021) build subgraphs by masking nodes and edges in the graph. They run their subgraph through the trained GNN and try to explain the differences to the entire graph with simple interpretable models and Shapley values. Zhang et al. (2021) infer subgraphs called prototypes that each represent one particular class. Graphs are classified and explained through their similarity to the prototypes.

**Example based.** Huang et al. (2020) proposes a graph version of the LIME (Ribeiro et al., 2016) algorithm. A prediction is explained through a linear decision boundary built by close-by examples. Vu & Thai (2020) aim to capture the dependencies in node predictions and express them in probabilistic graphical models. Faber et al. (2020) explain a node by giving examples of similar nodes with the same and different labels. Dai & Wang (2021) create a $k$-nearest neighbor model and measure similarity with GNNs. Yuan et al. (2020a) and Wang & Shen (2022) propose to generate a representative graph for each class in the dataset which maximize the models confidence in the class prediction.

While these methods provide some explanation, they do not grant true understanding of the processes defining each class. Gradient, Mutual-Information, and Counterfactual methods compute node-level importance that we can interpret as heatmaps. These heatmaps highlight what parts of the input are important but not why. Users might puzzle together a recipe by looking at heatmaps for many graph examples. However, this will be highly non-trivial. In order to understand the recipe of the enzymes example of Figure 1, a human would certainly need dozens of example graphs, and the correct thresholds (3,8) might still be out of reach.

Subgraph and Example based methods explain graphs by showing other example graphs or idealized prototype representatives for classes. Users might find a recipe by looking

at commonalities and differences for many graphs or prototypes, but again with considerable effort. In contrast, a recipe by GraphChef directly encodes the decision rules.

**Explanation properties and benchmarks.** Complimentary to the development of explanation methods is the research on how to evaluate these methods. Sanchez-Lengeling et al. (2020) and Yuan et al. (2020b) discuss desirable properties a good explanation method should have. For example, an explanation method should be faithful to the model, which means that an explanation method should reflect the model's performance and behavior. Agarwal et al. (2022) provide a theoretical framework to define how strong explanation methods adhere to these properties. They also derive bounds for several explanation methods. Faber et al. (2021) and Himmelhuber et al. (2021) discuss deficiencies in the existing benchmarks used for empirical evaluation. We will show recipes by GraphChef in Appendix A that exploit the flaws of these datasets.

**Simple GNNs.** Another interesting line of research is simplified GNN architectures (Cai & Wang, 2018; Chen et al., 2019; Huang et al., 2021). The main goal of this research is to show that simple architectures can perform competitively with traditional complex GNNs. As a side effect, the simplicity of these architectures also makes them slightly more understandable. However, they are not understandable to the extent that we can derive recipes for entire datasets.

### 2.2. Combining decision trees with neural networks

A recent work by Aytekin (2022) shows that we can transform any neural network into decision trees. However, this approach creates a tree with potentially exponentially many leaves. Even though this method produces decision trees we cannot use the outputs as humanly understandable recipes for datasets.

There is a successful line of work trying to distill neural networks into trees for better explainability (Boz, 2002; Craven & Shavlik, 1995; Dancey et al., 2004; Krishnan et al., 1999). More recently, Schaaf et al. (2019) have shown that encouraging sparsity and orthogonality in neural network weight matrices allows for model distillation into smaller trees with higher final accuracy. Wu et al. (2017a) follow a similar idea for time series data: they regularize the training process for recurrent neural networks to penalize weights that cannot be modeled easily by complex decision trees. Yang et al. (2018a) aim to directly learn neural trees. Their neural layers learn how to split the data and put it into bins. Stacking these layers creates trees. Kontschieder et al. (2015) learn neural decision forests by making the routing in nodes probabilistic and learning these probabilities and the leaf predictions.

GraphChef follows the same underlying idea. We want to structure a GNN in a way that allows for model distillation into decision trees to leverage their interpretability. In contrast to multi-layer perceptrons, we also have to reason about states in neighbors that are several hops away. Nevertheless, we believe that graphs and decision trees are a match made in heaven, since graph datasets often allow for short recipes.

## 3. The GraphChef Model

### 3.1. Creating a tree-based GNN Model

Our idea to create a fully-explainable model is to build a GNN which is composed of decision trees. We start our architecture from a GIN model (Xu et al., 2019) with $\epsilon = 0$. The GIN aggregation rule applies a parametrizable function $f_\theta$ on the node state and a sum over the node's neighbors' states:

$$h^{l+1}(v) = f_\theta(h^l(v), \sum_{w \in Nb(v)} h^l(w))$$

Additionally, our GIN model has an encoder layer for its input features and a decoder layer to map the final embeddings $h^L(v)$ to class probabilities using skip connections. For node classification, the decoder has access to the embeddings of each layer. For graph classification, the decoder has access to the pooled embeddings of each layer.

However, we cannot derive recipes for a dataset from a GIN model since the intermediate states $h^l(v)$ are continuous embeddings. This makes it hard to understand for humans what information is passed around between the nodes in each communication step. Loukas (2020) shows that GNNs such as GIN operate in a similar manner to synchronous message passing algorithms from distributed computing. Often, these algorithms have a limit on the message size of $b = O(\log n)$ bits (where $n$ is the number of nodes) but can perform arbitrary local computation (Peleg, 2000). In contrast to this, the stone age distributed computing model (Emek & Wattenhofer, 2013) assumes that each node uses a finite state machine to update its state and send its updated state as a message to all neighbors. The receiving node can count the number of neighbors in each state. A stone age node cannot even count arbitrarily, it can only count up to a predefined number, in the spirit of "one, two, many". Neighborhood counts above a threshold are indistinguishable from each other. Interestingly enough, such a simplified model can still solve many distributed computing problems (Emek & Wattenhofer, 2013).

We follow this model to build our dish (differentiable stoneh) layers. We extend the GIN update rule with a Gumbel-Softmax (Jang et al., 2016; Maddison et al., 2016) to produce categorical states. Formally:

$$h^{l+1}(v) = Gumbel(f_\theta(h^l(v), \sum_{w \in Nb(v)} h^l(w)))$$

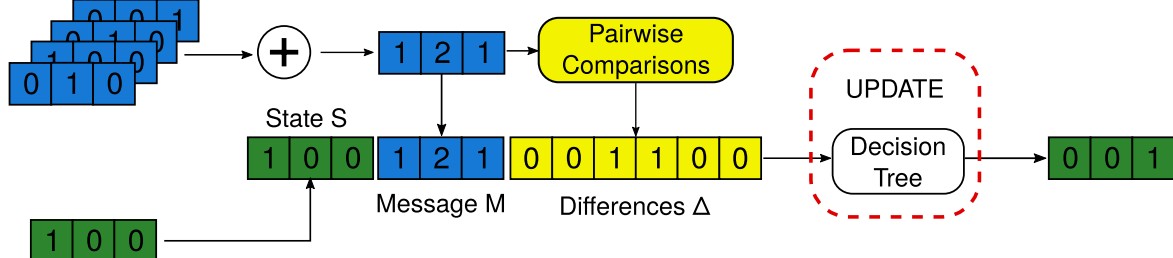

Figure 2: A GraphChef layer. GraphChef updates the state of a node based on its previous categorical state (here 0), the number of neighbors per state (1 in state 0, 2 in state 1, 1 in state 2), and binary $>$ comparisons between states (only state 1 outnumbers other states, therefore the third and fourth deltas are 1). A decision tree receiving this information computes the followup categorical state (here 2).

Furthermore, we also apply a Gumbel-Softmax to the result of the encoder layer. Therefore, hidden states become one-hot vectors and as a consequence the summation in the GIN update rule is now counting the number of neighbors in each state — just like in stone age. Unlike stone age, we did not find that limiting the counting gave better results or better interpretability.

From a theoretical perspective, the categorical state space of dish layers does not reduce expressiveness. If we have a GNN layer with continuous embeddings that encode $O(\log n)$ bits of information, we can construct a dish layer using $O(n)$ bits that can represent the same information. Practically, a state space with thousands of states is not tractable for a human and not interpretable. Therefore, we will constrain dish layers to few ($O(n)$) categorical states. In theory we look at an exponential loss in expressiveness. However, we noticed that in practice we incur hardly a loss for many datasets (c.f., Table 1a).

Next, we replace the update rule in each dish layer by a decision tree. We find these trees through model distillation: we pass all of the training graphs through the model and record all node states $h^l(v)$ in each layer. The tree for layer $l$ learns to predict $h^{l+1}(v)$ from $h^l(v)$ and $\sum_{w \in Nb(v)} h^l(w)$ over all nodes. Since node states are categorical in dish layers, this distillation is a classification problem. Formally our GraphChef layer computes:

$$h^{l+1}(v) = TREE^l(h^l(v), \sum_{w \in Nb(v)} h^l(w))$$

There is one caveat with using decision trees versus MLPs. Unlike MLPs, decision trees cannot easily compare two features, for example, to find out if one feature is larger than the other. To produce small trees we help the decision trees with comparisons: We include pairwise delta features $\Delta$, binary features that compare every pair of states. Let $c^l(v) = \sum_{w \in Nb(v)} h^l(w)$ be vector containing the numbers of neighbors of each state in layer $l$ and let S be the set of states. Then we compute the delta features as bits $b_{ij}$

between pairs of states if the number of neighbors in state $i$ outnumber the number of neighbors in state $j$. We compute these $Delta$ features for tree training and during inference. Figure 2 shows an example GraphChef layer that uses the following update rule:

$$\Delta(c^l) = \underset{i \in S, j \neq i \in S}{\|} \mathbb{1}_{c_i > c_j}$$
$$h^{l+1}(v) = TREE^l(h^l(v), c^l(v), \Delta(c^l(v)))$$

After training, we inspect the features that the tree uses. If the tree uses a feature from $h^l$, the tree considered the previous state of the node. We model this decision node as in Figure 3a. If the tree uses one of the features from $c^l$, the decision tree split is based on the neighborhood count of one particular state (shown in Figure 3b). The threshold $y$ is found by the decision tree during training. If tree uses one of the remaining features from $\Delta(c^l)$, the decision split is using one of the pairwise comparison features, which we can model as in Figure 3c. To obtain the final GraphChef model, we also distill the encoder and decoder layers to use decision trees.

### 3.2. Postprocessing GraphChef

**Pruning trees.** Decision trees like MLPs can theoretically be universal function approximators if they are sufficiently deep (Royden & Fitzpatrick, 1988). But again we prefer interpretable small and shallow decision trees. Shallow trees are more akin to the finite state machine used in the stone age distributed computing model. We limit the number of leaves in every tree during distillation but find that this alone is not sufficient. The decision trees still contain many nodes that are artifacts of overfitting to the training set.

We prune these nodes based on the reduced error pruning algorithm (Quinlan, 1987). First, we define a pruning set. We find that using only the validation set does not work. The validation set is too small to cover all paths in the trees which causes overpruning. On the other hand, we cannot use

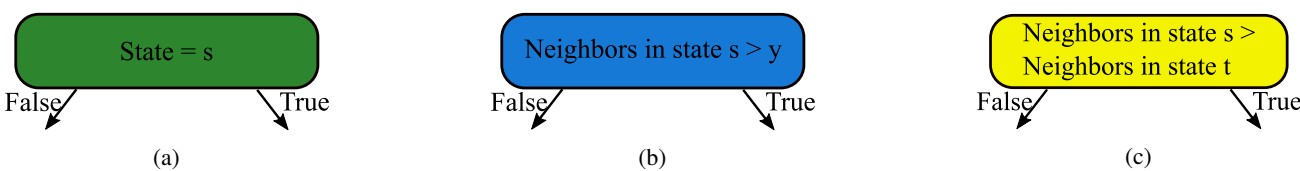

Figure 3: The different branches possible in a GraphChef layer. We can branch on (a) the state a node is in, (b) if the node has a certain number of neighbors in a certain state, or (c) if the node has more neighbors in one state than another state.

only the training set since since this set created the articates. We propose to use merge both sets for our pruning set.

For reduced error pruning we also need a quality criterion when we are allowed to replace an inner decision node with a leaf. We propose the following: If replacing a node with a leaf does not drop accuracy on the validation set, and does not drop training accuracy below validation accuracy, we accept the replacement. Not allowing validation accuracy to drop ensures that we do not overprune. But since we allow drops in training accuracy for the modified tree, we also remove decision nodes that result from overfitting. Not allowing training accuracy to drop below validation accuracy is another safeguard against overpruning. We keep iterating over all inner decision nodes, sorted by the number data points they cover, and try replacing them with leaf nodes until we find no more nodes to prune.

Empirically, we found that we can prune substantially more decision nodes when we allow for a slight deterioration in the validation accuracy. We follow the same approach as before and always prune the decision node that leads to the smallest deterioration. We repeat this until we are satisfied with the tradeoff between the model accuracy and tree size. Defining this tradeoff is difficult as it is subjective and specific to each dataset. Therefore, we included a feature in our user interface that allows users to try different pruning levels and report the impact on accuracy. These steps are incrementally pruning $10\%$ of the nodes in the losslessly pruned decision tree.

**Computing explanation scores.** If we have a recipe from GraphChef for the dataset, we can also compute heatmap-style importance scores for single graphs; similar to existing graph explanation methods. We compute these importance scores layer by layer. In the input layer, every node is its own explanation. In each GraphChef layer, we compute Tree-Shap values (Lundberg et al., 2018) for every decision tree feature. We then compute importance updates for every decision tree feature weighted by this Tree-Shap value independently (unused features have a value of $0$). If the node uses a state feature (as in Figure 3a) then we give importance to the node itself. If we use a message feature (as in Figure 3b), we distribute the importance evenly between all neighbors in this state. Last, if we use a delta feature (as

in Figure 3c), we distribute positive importance between all neighbors in the majority class and also negative importance between all neighbors in the minority class. Last, we normalize the scores back to sum up to $1$. In the decoder layer, we employ skip connections to also consider intermediate states (for node classification) or intermediate pooled node counts (for graph classification). We provide a detailed formal computation in Appendix D.

## 4. Experiments

### 4.1. Experiment setup

**Datasets.** We conduct experiments on two different types of datasets. First, we run GraphChef on synthetic GNN explanation benchmarks introduced by the previous works. We use the Infection and Negative Evidence benchmarks from Faber et al. (2021), The BA-Shapes, Tree-Cycle, and Tree-Grid benchmarks from Ying et al. (2019), and the BA-2Motifs dataset from Luo et al. (2020). We know ground truth recipes for these datastes so we can evaluate if the recipe found by GraphChef is correct. Secondly, we experiment with the following real-world datasets: MUTAG (Debnath et al., 1991); BBBP (Wu et al., 2017b); Mutagenicity (Kazius et al., 2005); PROTEINS, REDDIT-BINARY, IMDB-BINARY, and COLLAB (Borgwardt et al., 2005). We provide more information for all datasets, such as statistics, descriptions, and examples in Appendix E. Note that all datasets except COLLAB are small enough to train on a few commodity CPUs. For example, training the PROTEINS dataset for one seed on a laptop would take $5$ minutes for the full $1500$ epochs, a few seconds for the tree distillation, and $1 − 2$ minutes for tree pruning. The larger REDDIT-BINARY takes around one hour to train a dish GNN (if it uses all epochs) few seconds for distilling trees and around $10$ minutes for pruning. Computing lossy pruning takes a comparable amount of time to lossless pruning.

**Training and Hyperparameters.** We follow the same training scheme for all datasets similar to existing works (Xu et al., 2019). We do a $10−$fold cross-validation of the data with different splits and train both GraphChef and a baseline GIN architecture. The GNNs are trained on the training set for $1500$ epochs, allowing early stopping on the validation loss with patience of $100$. Each split uses early stopping on

| Dataset | GIN | dish GNN | GraphChef | | Hyperparameters | |
| | | | No pruning | Lossless pruning | Layers | States |
| --- | --- | --- | --- | --- | --- | --- |
| Infection | 0.98±0.04 | 1.00±0.00 | 1.00±0.00 | 1.00±0.00 | 5 | 6 |
| Negative | 1.00±0.00 | 1.00±0.00 | 1.00±0.00 | 1.00±0.00 | 1 | 3 |
| BA-Shapes | 0.97±0.02 | 1.00±0.01 | 0.99±0.01 | 0.99±0.01 | 5 | 5 |
| Tree-Cycles | 1.00±0.00 | 1.00±0.00 | 1.00±0.00 | 1.00±0.00 | 5 | 5 |
| Tree-Grid | 1.00±0.01 | 0.99±0.01 | 0.99±0.01 | 0.99±0.01 | 5 | 5 |
| BA-2Motifs | 1.00±0.00 | 1.00±0.00 | 1.00±0.00 | 1.00±0.00 | 4 | 6 |
| MUTAG | 0.88±0.05 | 0.88±0.06 | 0.88±0.06 | 0.85±0.08 | 4 | 6 |
| Mutagenicity | 0.81±0.02 | 0.79±0.02 | 0.75±0.02 | 0.74±0.02 | 3 | 8 |
| BBBP | 0.81±0.04 | 0.83±0.03 | 0.82±0.03 | 0.83±0.03 | 3 | 5 |
| PROTEINS | 0.70±0.03 | 0.71±0.02 | 0.71±0.04 | 0.71±0.04 | 3 | 5 |
| IMDB-B | 0.69±0.04 | 0.70±0.05 | 0.69±0.03 | 0.69±0.04 | 3 | 5 |
| REDDIT-B | 0.87±0.10 | 0.90±0.03 | 0.88±0.03 | 0.87±0.04 | 2 | 5 |
| COLLAB | 0.72±0.01 | 0.70±0.02 | 0.69±0.02 | 0.69±0.02 | 3 | 8 |

(a)                                        (b)

Table 1: a) Test set accuracies using different GNN layers. All methods perform virtually the same for all datasets. This shows that GraphChef layers match the expressiveness of GIN in practice. (b) The GraphChef hyperparameters are found through tree inspection.

the validation score. Both GNNs use a $2-$ layer MLP for the update function, with batch normalization (Ioffe & Szegedy, 2015) and ReLu (Nair & Hinton, 2010) in between the two linear layers. We use 5 layers of graph convolution. GIN uses a hidden dimension of 16, GraphChef uses a state space of 10. We also further divide the training set for GraphChef to keep a holdout set for pruning decision trees. After we train dish GNN with gradient descent, we distill the MLPs into decision trees. Each tree is limited to having a maximum of 100 nodes.

After training we can inspect the recipes from GraphChef and validate if all states and layers are necessary. If we find that the recipe uses less layers or states, we retrain with a smaller parameters set. Table 1b shows the number of layers and states that we find per dataset. A full model is used for GIN.

### 4.2. Quantitative Results

**GraphChef performs comparably to GIN.** First, we investigate the two assumptions that (i) A dish GNN matches the performance of GIN and (ii) that converting from a dish GNN to GraphChef also comes with little loss in accuracy. We further investigate how pruning impacts GraphChef's accuracy. In Table 1a we report the average test set accuracy for a GIN-based GNN, dish GNN, GraphChef with no pruning, and the lossless version of our pruning method.

We find that GraphChef performs almost identically to GIN.

This means that the understandable dataset recipes perform as well as the GNN. The model simplifications to obtain understandable recipes do not decrease accuracy. We observe, that tree pruning can even have a *positive* effect on test accuracy compared to non-pruned GraphChef. This is likely due to the regularization induced by the pruning procedure.

**GraphChef produces competitive explanations.** We further evaluate how good the explanations obtained through the recipes of GraphChef are. In line with previous work, we measure if GraphChef finds the nodes we know to be the correct explanation in the synthetic datasets Infection, Saturation, BA-Shapes, Tree-Cycles, and Tree-Grid. For example, we know that the correct explanation for a node in the Infection dataset is the shortest path from this node to an infected node. As another example, the correct explanation for every node in the house in the BA-Shapes dataset are all nodes in the house. Table 2 shows the explanation accuracy for GraphChef and baseline explanation methods. The importance scores gotten from GraphChef are competitive with explanation methods. We will show with the dataset recipes that the lower scores for Tree-Cycles and Tree-Grid are because of flaws in the dataset in Appendix A. These flaws were already identified by previous work (Faber et al., 2021; Himmelhuber et al., 2021).

**Pruning significantly reduces the decision tree sizes.** Third, we examine the effectiveness of our pruning method. We compare the tree sizes before pruning, after lossless

| Method | Infection | Saturation | BA-Shapes | Tree-Cycles | Tree-Grid |
|---|---|---|---|---|---|
| Gradient | 1.00±0.00 | 1.00±0.00 | 0.882 | 0.905 | 0.667 |
| GNNExplainer | 0.32±0.09 | 0.32±0.05 | 0.925 | 0.948 | 0.875 |
| PGMExplainer | 0.38±0.06 | 0.01±0.01 | 0.965 | 0.968 | 0.892 |
| GraphChef | 0.95±0.02 | 1.00±0.00 | 0.94±0.02 | 0.84±0.02 | 0.927±0.01 |

Table 2: Overlap of identified explanation to explanation ground truth. Numbers for GNNExplainer and PGMExplainer are taken from Ying et al. (2019), Vu & Thai (2020), and Faber et al. (2021).

| Dataset | No pruning | | REP Training | | REP Validation | | REP Ours | | REP Lossy | |
|---|---|---|---|---|---|---|---|---|---|---|
| | Accuracy | Size | Accuracy | Size | Accuracy | Size | Accuracy | Size | Accuracy | Size |
| Infection | 1.00±0.00 | 205±56 | 1.00±0.00 | 26±2 | 1.00±0.00 | 25±2 | 1.00±0.00 | 26±2 | 0.98±0.01 | 17±2 |
| Negative | 1.00±0.00 | 18±14 | 1.00±0.00 | 5±0 | 1.00±0.00 | 5±0 | 1.00±0.00 | 5±0 | 1.00±0.00 | 4±0 |
| BA-Shapes | 0.99±0.01 | 30±10 | 0.99±0.01 | 21±5 | 0.97±0.03 | 15±4 | 0.99±0.01 | 21±5 | 0.98±0.04 | 17±4 |
| Tree-Cycles | 1.00±0.00 | 19±5 | 1.00±0.00 | 11±3 | 0.99±0.02 | 9±2 | 1.00±0.00 | 11±3 | 0.99±0.01 | 9±3 |
| Tree-Grid | 0.99±0.01 | 30±13 | 0.99±0.01 | 17±8 | 0.99±0.01 | 13±4 | 0.99±0.01 | 15±8 | 0.99±0.01 | 15±8 |
| BA-2Motifs | 1.00±0.00 | 141±43 | 1.00±0.00 | 12±3 | 1.00±0.01 | 11±3 | 1.00±0.00 | 13±4 | 1.00±0.00 | 13±4 |
| MUTAG | 0.88±0.06 | 59±27 | 0.86±0.08 | 19±17 | 0.83±0.07 | 7±6 | 0.85±0.08 | 18±16 | 0.85±0.08 | 18±16 |
| Mutagenicity | 0.75±0.02 | 375±13 | 0.76±0.02 | 154±19 | 0.73±0.01 | 56±16 | 0.74±0.02 | 91±36 | 0.73±0.02 | 50±19 |
| BBBP | 0.82±0.03 | 366±53 | 0.84±0.02 | 88±52 | 0.79±0.04 | 8±10 | 0.83±0.03 | 46±27 | 0.82±0.03 | 31±18 |
| PROTEINS | 0.71±0.04 | 206±90 | 0.72±0.03 | 12±13 | 0.70±0.04 | 8±6 | 0.71±0.04 | 9±6 | 0.71±0.04 | 9±6 |
| IMDB-B | 0.69±0.03 | 218±32 | 0.69±0.04 | 20±9 | 0.66±0.06 | 16±6 | 0.69±0.04 | 29±9 | 0.69±0.04 | 29±9 |
| REDDIT-B | 0.88±0.03 | 248±28 | 0.88±0.02 | 53±14 | 0.85±0.04 | 28±8 | 0.87±0.04 | 49±21 | 0.87±0.04 | 38±15 |
| COLLAB | 0.69±0.02 | 301±1 | 0.70±0.02 | 36±15 | 0.67±0.03 | 22±12 | 0.69±0.02 | 30±18 | 0.68±0.02 | 21±12 |

Table 3: Running reduced error pruning (REP) on different pruning sets. Lossy pruning prunes the nodes with least loss in accuracy up to a manually chosen threshold.

pruning, and after lossy pruning. We measure tree size as the sum of decision nodes over all trees. Additionally, we verify the effectiveness of using our pruning criterion for reduced error pruning and compare it against simpler setups of using only the training or validation set for pruning. We report the tree sizes and test set accuracy for all configurations in Table 3.

We can see that the reduced error pruning leads to an impressive drop in the number of nodes required in the decision trees. On average, we can prune about 62% of nodes in synthetic datasets and even around 84% of nodes in real-world datasets without a loss in accuracy. If we accept small drops, in accuracy we can even prune a total of 68% and 87% of nodes in synthetic and real-world datasets, respectively. If we compare the different approaches for reduced error pruning, we can see that our proposed approach of using both training and validation accuracy performs the best. As expected, pruning only on the validation set tends to over-prune the trees: Trees become even smaller but there is also a larger drop in accuracy, especially in the real-world datasets. Using the training set leads to underpruning, there is no drop in accuracy but decision trees for real-world graphs tend to stay large.

### 4.3. Qualitative Results

Last, let us look at one example how to parse a recipe from GraphChef that is given by its decision trees. We analyze GraphChef for the Reddit-Binary dataset. The recipe is shown in Figure 4. First, we will create an understanding for every categorical state in every layer. We can do so in an iterative fashion: We first understand the encoder layers, then use these interpreations to understand the first layer, then use these interpretations for the second layer . . . . We provide this table for the Reddit-Binary dataset in Table 4. To understand the dataset, we can inspect the decoder rules.

We can understand that we need to find a certain amount of users (15) fulfilling certain conditions: 1) inactive users writing with at least two central users or 2) active or central users that write with at least one central user and at most 2 active useres. We can understand 1) as just replying to a single central user is not sufficient. We hypothesize that also controversial opinions in discussions can attract many comments, even by inactive users. Users fulfilling 2) write mostly with inactive users since there are few central users and few communication with active users is allowed.

GraphChef's recipe aligns with our belief that Q/A graphs

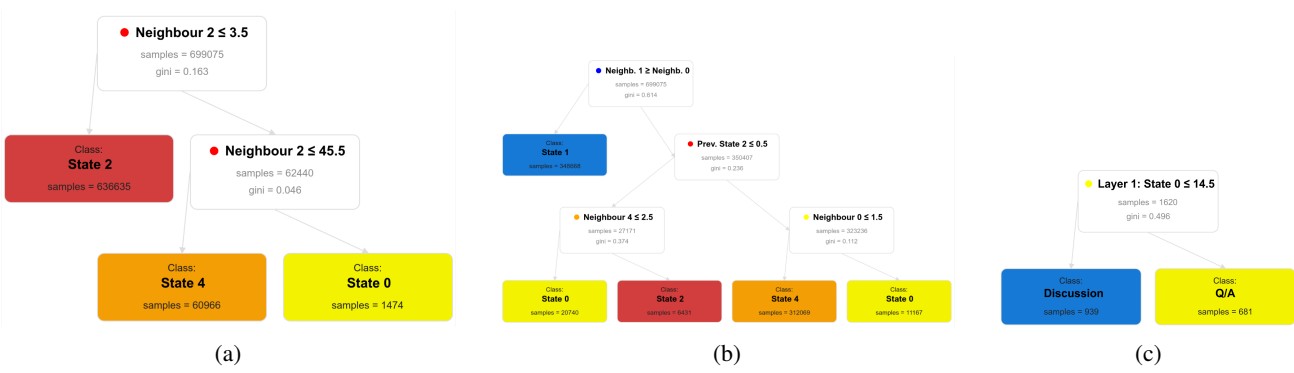

Figure 4: GraphChef recipe for the Reddit-Binary dataset. Table 4 provides an interpretation of all states in all layer and for the entire dataset.

| Layer | State | Decision Rule | Interpretation |
|---|---|---|---|
| Encoder | 2 | All nodes | No differentiation due to no features. |
| Layer 0 | 2 | Nodes with at most 3 neighbors | Inactive users |
| Layer 0 | 4 | Between 4 and 45 neighbors | Active users |
| Layer 0 | 0 | More than 45 neighbors | Central users |
| Layer 1 | 0 | 1) State 0 nodes with more than one state 0 neighbor or 2) Not state 2 nodes that have at most two state 4 neighbors | 1) Inactive users writing with at least 2 central users or 2) Active or central users that write with at most 2 active users. |
| Layer 1 | 1 | No neighbor in state 0 | Users that do no write with a central user. |
| Layer 1 | 2 | Not state 2 nodes with at least 3 state 4 neighbors | Active or central users that write with at least 3 active users. |
| Layer 1 | 4 | Nodes in state 2 with exactly one state 0 neighbor | Inactive users write with one central user. |
| Decoder | Q/A | At least 15 nodes in Layer 1 state 0 | See interpretation of Layer 1 state 0. |
| Decoder | Discussion | Otherwise | The GraphChef model looks for evidence of a Q/A graph. Discussions are "not Q/A" graphs. |

Table 4: Analysis of GraphChef recipe in Figure 4 for the Reddit-Binary dataset. A Q/A graph requires central users and 15 users that 1) are inactive and write with more than one central user or 2) are active or central and write mostly with inactive users.

are more "star-like" than discussion graphs. Unlike previous methods (Faber et al., 2020), the recipe defines what we should consider as star like. We obtain a threshold what we can consider as centers of the star (central nodes with a degree of 46 or higher). Furthermore, the recipe tells us to what extent non-star communication is acceptable for a Q/A graph (two active users). To the best of our knowledge such insight about the Reddit-Binary dataset has not been found yet with existing explanation methods.

We also provide the recipes of GraphChef with analysis for MUTAG, BA-2Motifs, BA-Shapes, Tree-Cycle, and Tree-Grid in Appendix A. We verify these recipes against the ground truth of the datasets. GraphChef also reveals and exploits some dataset flaws that Faber et al. (2021); Himmelhuber et al. (2021) analyzed before. For example, GraphChef uses the bias trick for the BA-2Motifs dataset.

## 5. Conclusion

In this paper, we presented GraphChef, the first architecture that can compute abstract rules to describe entire graph datasets, and not just single graphs. We believe that GraphChef will help improve our understanding of graph problems. This will be crucial in the adoption of graph learning models in security critical domains such as medicine. We can also identify and reject recipes that make biased or discriminatory decisions. As shown with MUTAG, PROTEINS, or Reddit-Binary, we hope that GraphChef's recipes help experts in various domains to better understand the rules behind their data. As a limitation, we observe that GraphChef struggles with datasets that have thousands of node input features, e.g. Cora (Sen et al., 2008). We discuss this issue in more detail in Appendix B.

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

# A. More Graphchef Recipe Analyses

## A.1. MUTAG

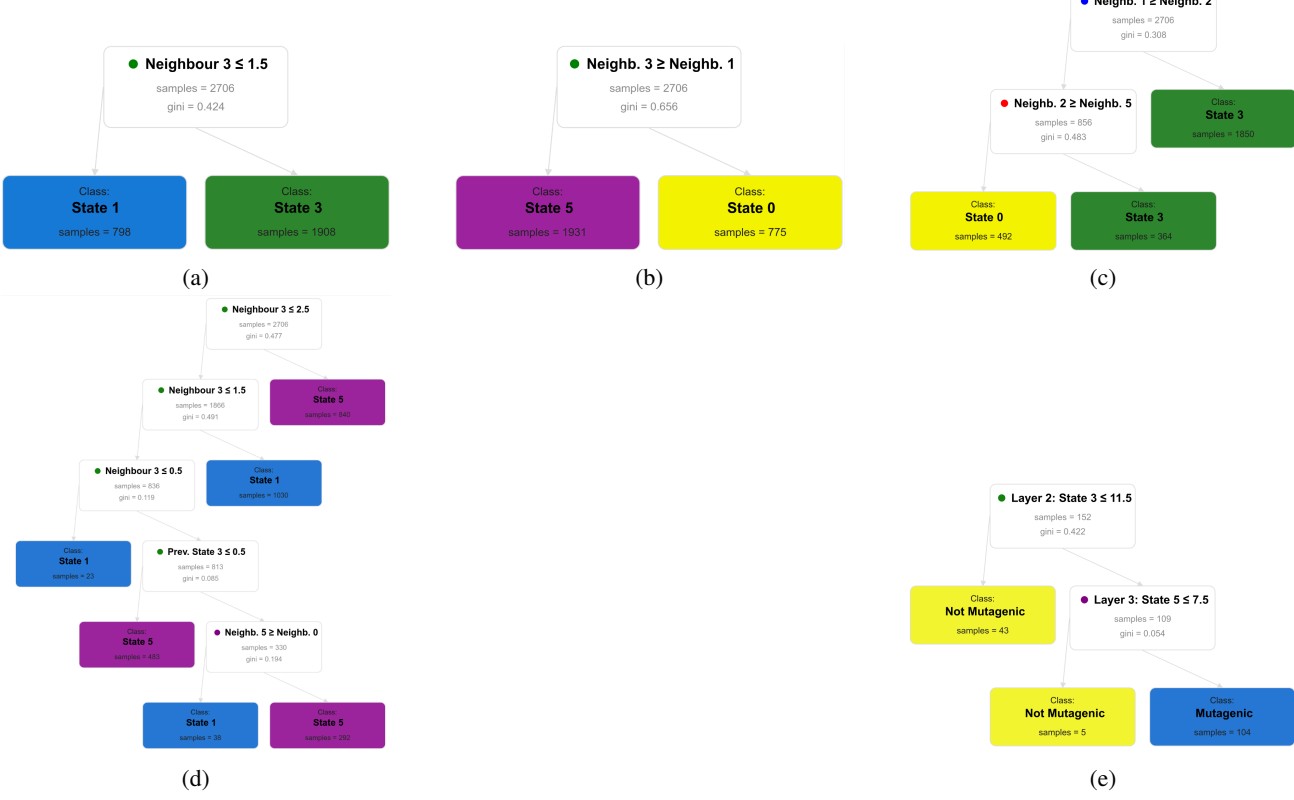

Figure 5: GraphChef recipe for MUTAG. Table 5 shows an interpretation for all states in all layers. For a graph to be mutagenic, it requires at least twelve atoms other than O and eight atoms that 1) have three or more non-O bindings 2) are O atoms, 3) bound to $O_2$ atoms.

Table 5 shows an incremental interpretation (from Encoder to Decoder) of the states in all layers that are shown in the trees the GraphChef recipe in Figure 5 for the MUTAG dataset. The encoder shows that graph size is important (we need at least twelve non-O atoms) and O atoms and their connectivity play a role. We need at least eight nodes that 1) have three non-O bindings 2) are O atoms 3) are connected to $O_2$ groups. The last two conditions highlight why $NO_2$ groups are associated with mutagenicty: The N atom fulfills the last condition and the two O atoms fulfill the other condition. However, in the MUTAG (Debnath et al., 1991) dataset, these structures are not sufficient for mutagenic molecules.

| Layer | State | Decision Rule | Interpretation |
|---|---|---|---|
| Encoder | 3 | All nodes receive state 3 | GraphChef drops atom types (and rediscovers them later via degrees). |
| Layer 0 | Less than two neighbors | Degree 1 nodes, H atoms are implicit so these represent O atoms. | |
| Layer 0 | 3 | At least two neighbors | Nodes with at least two electron bindings, predominantly C and N. |
| Layer 1 | 0 | More neighbors in state 1 than state 3 | Atoms with majorly bindings to O atoms. |
| Layer 1 | 5 | At least as many state 3 as state 1 neighbors | Atoms not connected to O, or mainly to other atom types. |
| Layer 2 | 0 | No neighbor in state 5 | Rediscovers almost all O atoms, especially those in $O_2$ groups. They have one neighbor each and that neighbor has bindings to mostly O. |
| Layer 2 | 3 | At least one neighbor in state 5 | Atoms other than O and some O atmons that are not in $O_2$ groups. |
| Layer 3 | 1 | 1) Exactly two state 3 neighbors 2) Nodes with no state 3 neighbors 3) Nodes with 1 state 3 neighbor but at most state 0 neighbor | 1+3) Nodes with at most one O neighbor and at most two other neighbors 2) Nodes with only O neighbors. |
| Layer 3 | 5 | 1) Nodes with at least 3 state 3 neighbors 2) not state 3 nodes 3) state 3 nodes with one state 3 and 2 state 0 neighbors | 1) Atoms with at least 3 connections to atoms other than O 2) O atmons 3) atoms connected to $O_2$ groups. |
| Decoder | Mutagenic | At least twelve atoms in layer 2 state 3 and at least eight nodesin layer 3 state 5. | At least twelve atoms other than O and 1) Atoms with at least 3 connections to atoms other than O 2) O atmons 3) atoms connected to $O_2$ groups. |
| Decoder | Not Mutagenic | otherwise | otherwise |

Table 5: Analysis of the GraphChef recipe in Figure 5 for the MUTAG dataset. For a graph to be mutagenic, it requires at least twelve atoms other than O and eight atoms that 1) have three or more non-O bindings 2) are O atoms, 3) bound to $O_2$ atoms.

## A.2. BA-2Motifs

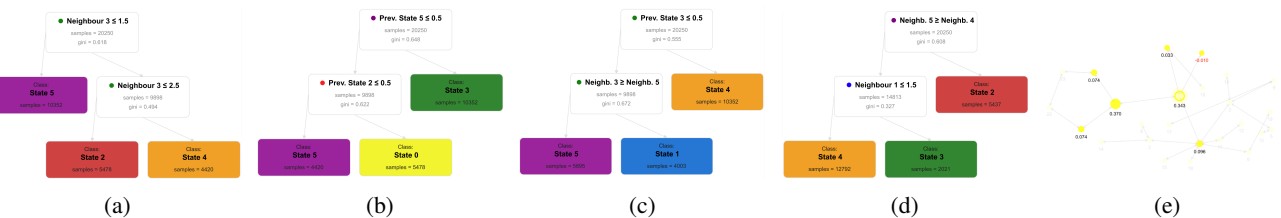

(a)  (b)  (c)  (d)  (e)

Figure 6: Layers of GraphChef for the decision process on BA-2MOTIFS. Table 6 shows an interpretation for all states in all layers. The model learns to identify house nodes and classify such graphs. Cycle graphs are graphs which are not house graphs, thus solved with the bias term. Explanation scores for cycles are therefore off (e).

| Layer | State | Decision Rule | Interpretation |
|---|---|---|---|
| Encoder | 3 | All nodes | No node features available for differentiation. |
| Layer 0 | 2 | Two state 3 neighbors | Degree 2 nodes. |
| Layer 0 | 4 | Three or more state 3 neighbors | Degree 3 or higher nodes. |
| Layer 0 | 5 | Less than two neighbors | Degree 1 nodes (graphs are connected). |
| Layer 1 | 0 | State 2 nodes | Degree 2 nodes. |
| Layer 1 | 3 | State 5 nodes | Degree 1 nodes. |
| Layer 1 | 5 | Neither state 2 nor 5 | Degree 3 or higher nodes. |
| Layer 2 | 1 | Not state 3 nodes with more state 5 than state 3 neighbors. | House candidates: nodes with a degree of at least 2, with at least one degree 3 or higher neighbor and no degree 1 neighbor. |
| Layer 2 | 4 | State 3 nodes | Degree 1 nodes. |
| Layer 2 | 5 | Not state 3 nodes with state 5 neighbors | At least degree 2 nodes but connected to at least degree 1 neighbor (which house nodes do not have). |
| Layer 3 | 2 | More state 4 than state 5 neighbors | Nodes that have majorly degree 1 neighbors (not house nodes). |
| Layer 3 | 3 | At least two state 1 neighbors | House nodes: connected to two more house candidates. |
| Layer 3 | 4 | At most one state 1 neighbor | Nodes connected to at most one house candidate (wrong for every node in the house). |
| Decoder | House | At least five nodes in layer 3 state 3. | Graphs with at least five house nodes. |
| Decoder | Cycle | otherwise | otherwise. |

Table 6: Analysis of the GraphChef recipe in Figure 6 for the BA-Motifs dataset. The model learns to identify house nodes and classify such graphs. Cycle graphs are graphs which are not house graphs, thus solved with the bias term.

Table 6 shows an interpretations of the states in all layers for the GraphChef recipe for BA-2MOTIFS. Figure 5 shows the recipe. GraphChef only learns to identify house nodes. The important step is state 1 in the second layer. Due to the Barabasi-Alert base graph structure, house nodes stand out with their degree of 2 or 3. The next layer confirms the house as house candidates that are connected to house candiates. The model does not learn about cycles at all, Cycles are "not houses". We can see that GraphChef found and exploited the pitfall about bias terms noted by Faber et al. (2020); Himmelhuber et al. (2021). Figure 6e shows that we cannot trust explanation scores for cycles.

## A.3. Tree Cycle

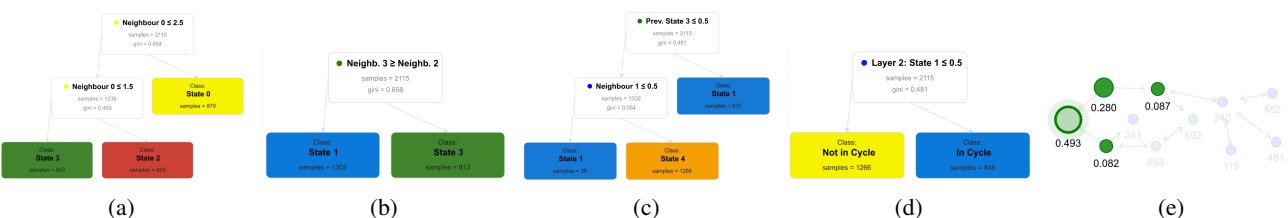

Figure 7: GraphChef recipe for TREE-CYCLE. Table 6 shows an interpretation for all states in all layers. A degree check for degree 2 nodes find the cycles quickly without considering the whole structure. Explanation scores for cycles are therefore off (e).

| Layer | State | Decision Rule | Interpretation |
|---|---|---|---|
| Encoder | 0 | All nodes | No node features available for differentiation. |
| Layer 0 | 0 | Three or more state 0 neighbors | Degree 3 or higher nodes (inner nodes in the tree, cycle node connecting the cycle to the tree). |
| Layer 0 | 2 | Two neighbors in state 0 | Degree 2 nodes (root node and cycles nodes). |
| Layer 0 | 3 | One or zero neighbors in state 0 | Degree 1 nodes (leaves in the connected graph). |
| Layer 1 | 1 | At least as many state 3 as state 2 neighbors | As least as many leaves as cycle neighbors (true for inner nodes as well having zero of both). |
| Layer 1 | 3 | More state 2 than state 3 neighbors | Most cycle nodes, nodes connected to degree 2 nodes. |
| Layer 2 | 4 | Not previous state 3 and at least one state 1 neighbor | Not already a cycle node and connected to a non-cycle node. |
| Layer 2 | 1 | 1) Previous state 3 or 2) no state 1 neighbors | 1) already a cycle node or 2) only connected to cycle nodes. |
| Decoder | Cycle | In layer 2 state 1 | See previous state. |
| Decoder | No Cycle | otherwise | otherwise. |

Table 7: Analysis of the GraphChef recipe in Figure 7 for the Tree-Cycle dataset. The base graph contains only one degree two node that is not part of a cycle. A degree check quickly finds the cycles.

Table 7 shows an interpretations of the states in all layers for the GraphChef recipe for BA-2Motifs. Figure 7 shows the recipe. Due to the base graph being a binary tree, degree two nodes (especially those connected to degree two nodes) are a strong indicator for cycles. For almost all nodes, GraphChef can identify if they are part of the cycle after two layers. Therefore, we do not even need the whole cycle. This is consistent with previous analysis on Tree-Cycle that not the whole cycle is necessary (Faber et al., 2020; Himmelhuber et al., 2021). Figure 7e shows that we cannot trust explanation scores for cycles since they find only a subset of the motif.

## A.4. Tree Grid

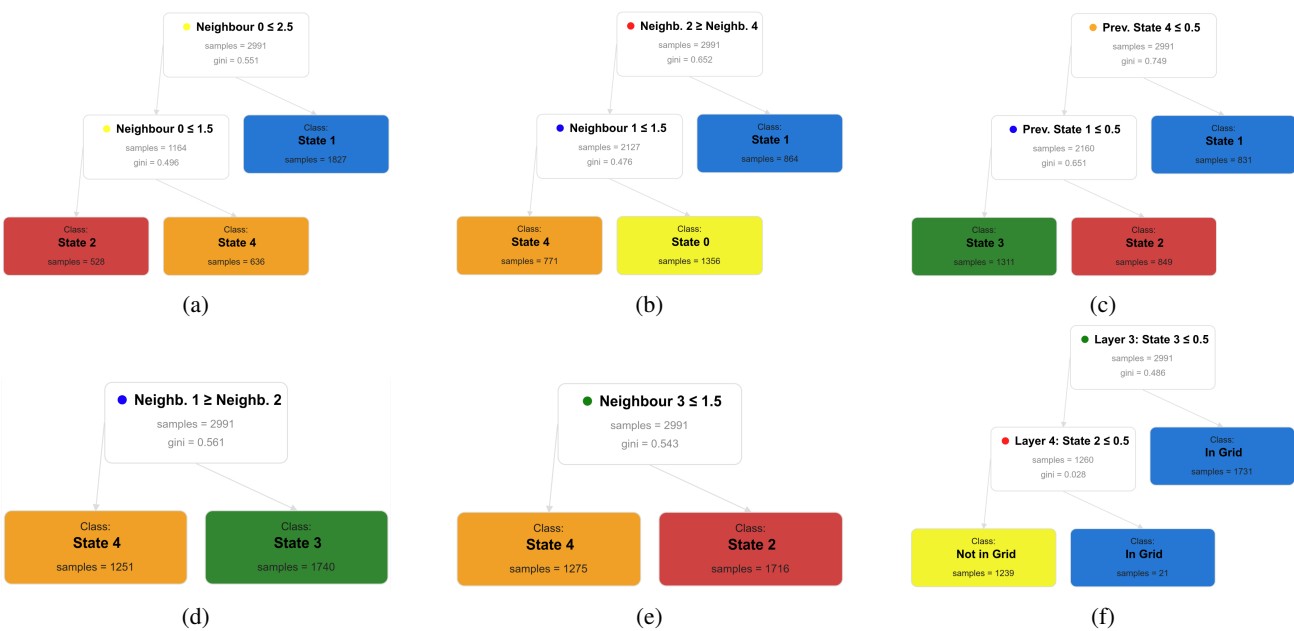

Figure 8: GraphChef recipe for TREE-GRID. Table 8 shows an interpretation for all states in all layers. A degree check for degree 2 nodes find the corner nodes in the grid, after which we explore the remaining motif.

The Tree-Grid (Ying et al., 2019) dataset is similar to the Tree-Cycles dataset we discussed in the main body of the paper. The base graph is a balanced binary tree to which we append $3 \times 3$ grids. As in the Tree-Cycles example, there are (apart from the root node) no other nodes with degree 2 which makes bootstrapping the grid discovery easier. As in the Tree-Cycles example, a GNN does not need to see the whole grid to make a prediction. Table 8 shows the interpretation of the layers and states of GraphChef shown in Figure A.4. The corner nodes in the grid can be quickly found with a degree check. Then the remaining grid can be found by exploring the neighborhood.

Some nodes can identify that they are part of the grid in just three layers — importantly the corner nodes generally belong to this group. This means these nodes need not consider the opposite corner node at distance $4$. GraphChef does not include this node in its explanation. This is consistent with the explanation accuracy in Table 2: GraphChef achieves a bit more than $\frac{8}{9}$ which means that one node is missing.

| Layer | State | Decision Rule | Interpretation |
|---|---|---|---|
| Encoder | 0 | All nodes | No node features available for differentiation. |
| Layer 0 | 1 | Three or more state 0 neighbors | Degree 3 or higher nodes. (Inner tree nodes and grid nodes except corners) |
| Layer 0 | 2 | Less than two state 0 neighbors | Degree 1 nodes (leaves in the tree). |
| Layer 0 | 4 | Two state 0 neighbors | Degree 2 nodes (root node, corner nodes in the grid). |
| Layer 1 | 0 | At least as many state 2 as state 4 neighbors and at least two state 1 neighbors | Inner nodes in the tree and grid, except parents of leaf nodes. |
| Layer 1 | 1 | More state 4 than state 2 neighbors | Nodes connected to the root or grid corners. |
| Layer 1 | 4 | At least as many state 2 as state 4 neighbors and at most one state 1 neighbors | Leaves and their parent nodes. |
| Layer 2 | 1 | Previous state 4 nodes | Leaves and their parents. |
| Layer 2 | 2 | Previous state 1 nodes | Nodes connected to the root or grid corners. |
| Layer 2 | 3 | Otherwise | Inner nodes in the tree and grid, except parents of leaf nodes. |
| Layer 3 | 3 | More state 2 than state 1 neighbors | More corner and root nodes in the distance 2 (can be the node itself) than leaves. This captures corner and center nodes in the grid |
| Layer 3 | 4 | At least as many state 1 as state 2 neighbors. | Most nodes since they are connected to leaves or not to the root. |
| Layer 4 | 2 | At least two state 3 neighbors | Nodes with two grid neighbors, finding the remaining grid nodes. |
| Layer 4 | 4 | At most one state 3 neighbor | Nodes with at most one grid neighbor. |
| Decoder | Grid | Layer 3 state 3 or layer 4 state 2 | Corner or center grid node or nodes connected to two such nodes. |

Table 8: Analysis of the GraphChef recipe in Figure A.4 for the Tree-Grid dataset. The base graph contains only one degree two node that is not part of a cycle. A degree check for two quickly finds the corner nodes of grids as a starting point to discover the motif.

## A.5. BA-Shapes

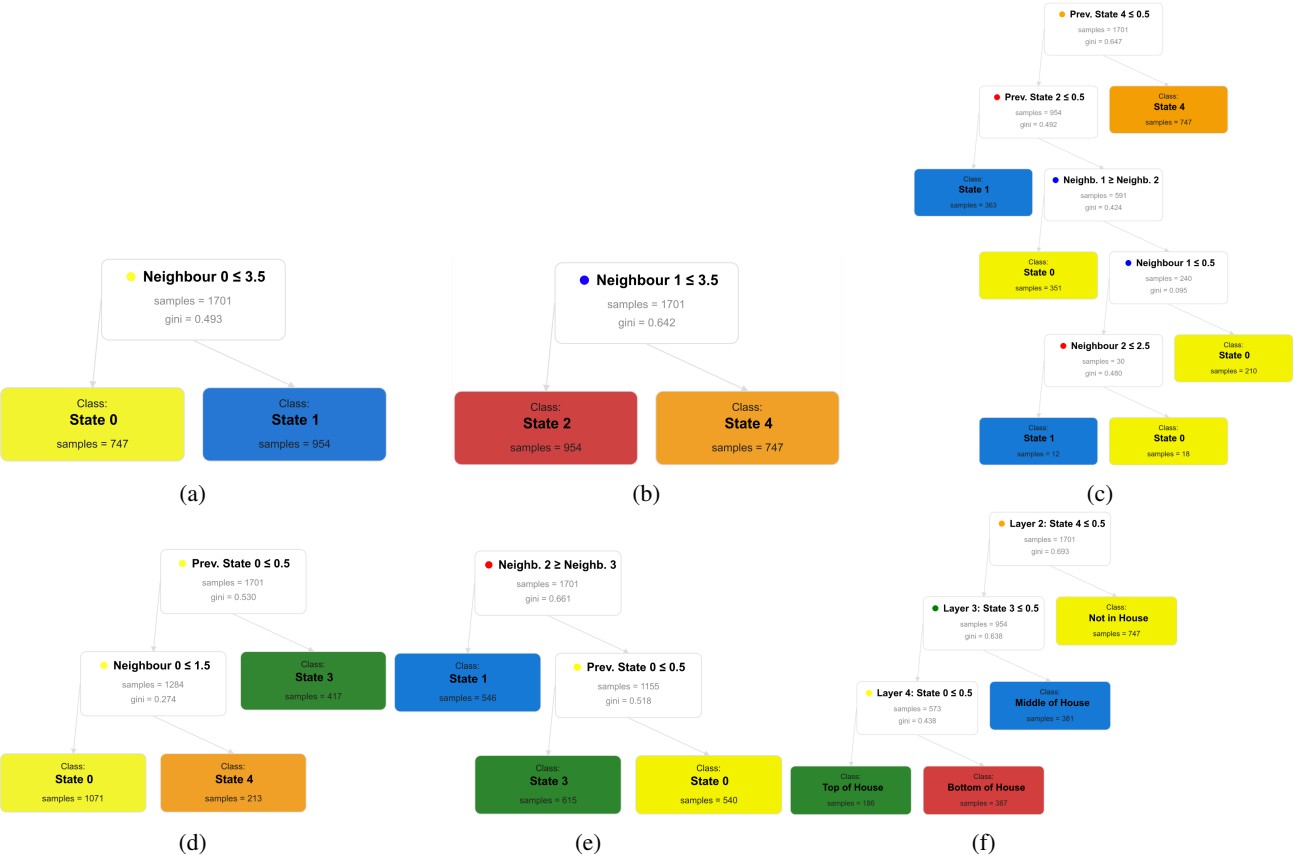

Figure 9: GraphChef recipe for BA-SHAPES. Table 8 shows an interpretation for all states in all layers. Degrees below 4 connected to more nodes with such degrees finds the general house. Middle nodes are degree 3 in this structure; Top nodes are connected to both middle nodes; Bottom nodes are connected to one.

Let us now look at the BA-SHAPES dataset to classify each node with its position in a house motif, or if the node is part of the Barabasi-Albert base graph. Nodes in the base graph generally have a high degree, we can find house nodes as being connected to almost no high-degree nodes but only among each other. From there we subdivide house nodes: Middle nodes have a 3 house neighbors, the top node is connected to both middle nodes, and the bottom nodes only to one.

| Layer | State | Decision Rule | Interpretation |
|---|---|---|---|
| Encoder | 0 | All nodes | No node features available for differentiation. |
| Layer 0 | 0 | Less than four state 0 neighbors | Degree 3 or lower nodes. (lower degree) |
| Layer 0 | 1 | At least four state 0 neighbors | Degree 4 or higher nodes. (high degree) |
| Layer 1 | 2 | At most 3 neighbors of state 1 | At most three high-degree neighbors. (House candidates, all nodes in the house have lower degree, the basegraph has many high-degree nodes). |
| Layer 1 | 4 | At least four neighbors in state 1 | At least four high-degree neighbors. |
| Layer 2 | 0 | Three state 2 neighbors | Three house candidates in the neighborhood (middle nodes in the house). |
| Layer 2 | 1 | At most state 2 neighbors | Top or bottom of the house. |
| Layer 2 | 4 | Previous state 4 nodes | At least four high-degree neighbors. |
| Layer 3 | 0 | At most one state 3 neighbor | Nodes connected to zero or one middle house node. |
| Layer 3 | 3 | Previous state 0 | Middle house nodes. |
| Layer 3 | 4 | At least two state 0 neighbors | Top house nodes. |
| Layer 4 | 0 | At least one state 3 neighbor and previous state 0 | Bottom house node: nodes connected to one house middle node. |
| Layer 4 | 1 | No state 3 neighbors | Nodes not connected to the house. |
| Layer 4 | 2 | At least one state 3 neighbor but not previous state 0 | Top node in the house. |
| Decoder | Not in house | Layer 2 state 4 | At least four high-degree neighbors. |
| Decoder | Middle of House | Not above and layer 3 state 3 | Three house candidates in the neighborhood. |
| Decoder | Bottom of House | Not above and layer 4 state 0 | Nodes connected to one house middle node. |
| Decoder | Top of House | Otherwise (but GraphChef could have used layer 3 state 4) | Otherwise (or nodes connected to two house middle nodes). |

Table 9: Analysis of the GraphChef recipe in Figure A.5 for the BA-Shapes dataset. The base graph has many edges, house nodes stand out by having a degree of 3 or lower and being connected to such nodes.

## B. GraphChef on datasets with many input features

| Dataset | Features | GIN | GraphChef | | |
|---|---|---|---|---|---|
| | | | Differentiable | No pruning | Lossless pruning |
| CORA | 1433 | 0.87±0.02 | 0.82±0.03 | 0.69±0.04 | 0.68±0.03 |
| CiteSeer | 3703 | 0.77±0.01 | 0.70±0.03 | 0.61±0.04 | 0.61±0.02 |
| PubMed | 500 | 0.88±0.01 | 0.87±0.01 | 0.85±0.01 | 0.85±0.01 |
| OBGN-Arxiv | 128 | 0.68±0.02* | 0.68±0.01 | 0.28±0.11 | - |

Table 10: GraphChef results for citation datasets with high-degree counts. *Since the dataset has 40 classes, we use a state-size of 50 for GraphChef variants and 128 wide embeddings for GIN.

In the following we want to discuss GraphChef on high-dimensonal datasets such as Cora (1433) features. Table 10 shows a comparison of GIN, dish GNN and GraphChef similar to Table 1a. The results are mixed: on Pubmed, GraphChef performs comparable to GIN, on Cora there is a small drop for dish GNN but a significant drop when converting to trees. For CiteSeer, both dish GNN and converting to trees cause clear drops in accuracy. We see two factors that make this dataset challenging: Large feature spaces make it harder to reduce to a categorical state. For example for the Cora dataset, the encoder needs to reduce from 1433 to 10 features. This effect increases in GraphChef when we limit the number of leaves: Having 100 decision leaves means that a tree can have 99 decision nodes and look at most at 99 features. But already such trees are impractical to interpret. We found that even after pruning, the trees often contain long paths of depth 20 or more. The problems aggravate on the larger OBGN-Arxiv dataset: dish GNN performs decently with a drop comparable to CiteSeer but GraphChef drops drastically in accuracy. Furthermore, this dataset reveals scalability limits for GraphChef's pruning method: Pruning requires the number of leaves squared many runs over the dataset and does not scale to this dataset.

Therefore we believe that handling such datasets requires a different approach. In future work, we image that these issues could be addressed through approaches such as PCA, clustering, or special MLP construction techniques (Wu et al., 2017a; Schaaf et al., 2019) to reduce the input space without breaking the interpretability chain before applying GraphChef.

# C. Using the tool

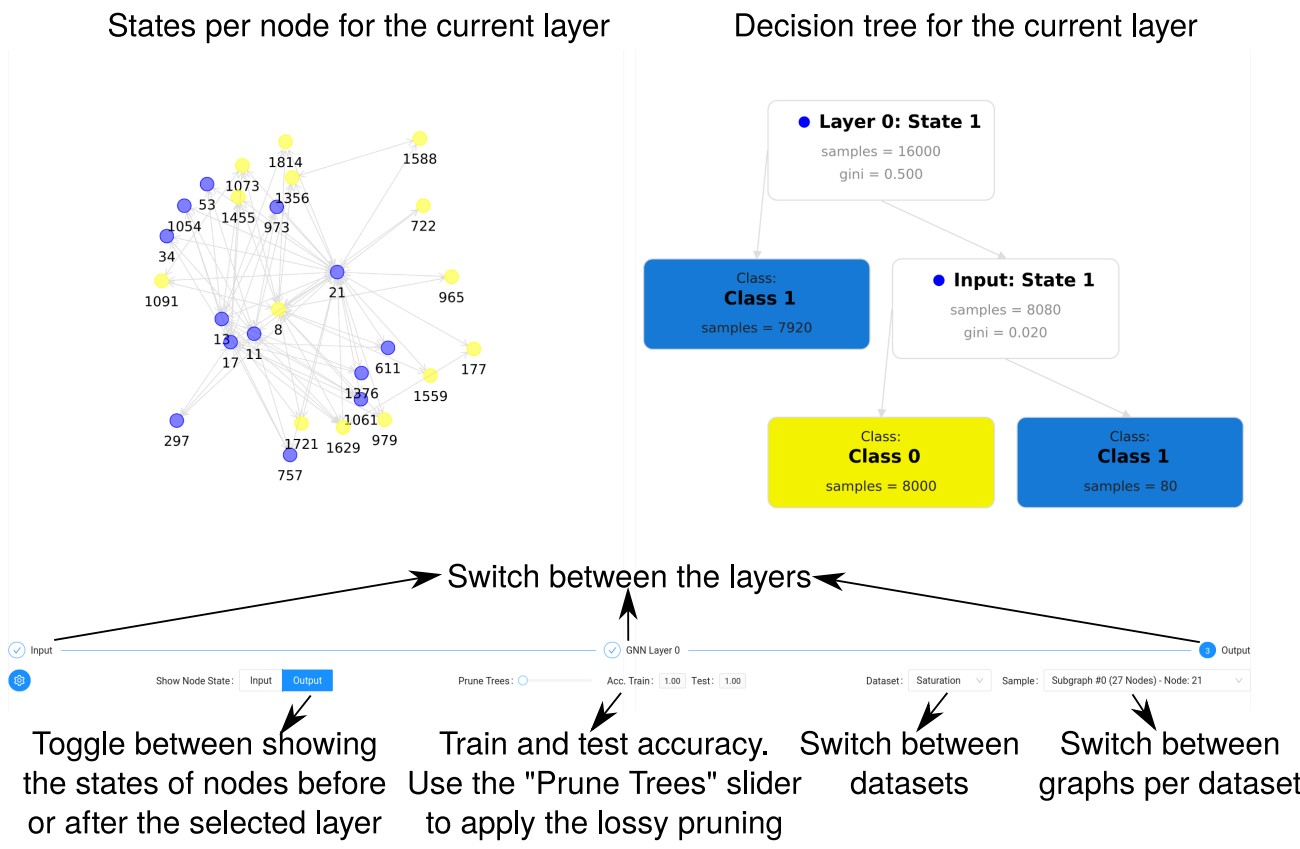

Figure 10: Initial page for the web tool. We can see the decision trees for GraphChef per dataset and which node for a graph is in what state. We can switch layers, graphs and datasets. We can also see the test accuracy for the current setting and choose an amount of lossy pruning. with the slider.

A example instance of the tool is deployed and available via Netlify[2] and can be accessed under the link `https://interpretable-gnn.netlify.app/`. The supplementary material also contains code to host the interface yourself, in case you want to try variations of GraphChef. In the backend, we use PyTorch (Paszke et al., 2019)[3] and PyTorch Geometric (Fey & Lenssen, 2019)[4] to train GraphChef and SKLearn(Pedregosa et al., 2011)[5] to train the decision trees.

The tool is built with React, in particular the Ant Design library.[6] We visualize graphs with the Graphin library.[7] The interface is a single page that will look similar to Figure 10.

The largest part of the interface is taken by two different panels at the top. In the right panel, you can see the decision tree for the currently selected layer. The trees use the three branching options from Figure 3. In the interface, evaluating the branching to true means taking the left path (this is opposite to Figure 3, which we will flip). In the left panel, you can see an example graph and which nodes end up in which state after this layer (in the bottom left you can toggle to see the input states instead). This panel does not show the full graph (most graphs in the datasets are prohibitively large) but an excerpt around an interesting region. Directly below these two graphics, you have the option to switch between layers by clicking on the respective bubble.

---

[2]`https://netlify.com/`

[3]`https://github.com/pytorch/pytorch`

[4]`https://github.com/pyg-team/pytorch_geometric`

[5]`https://github.com/scikit-learn/scikit-learn`

[6]`https://github.com/ant-design/ant-design/`

[7]`https://github.com/antvis/Graphin`

In the bottom right, you can switch to a different graph in the same dataset or to a different dataset. In the centre, you can see the accuracy of GraphChef with the displayed layers. The slider allows to apply the lossy pruning from Section 3.2 and the accuracy values update to the selected pruning level.

The interface also allows us to examine a single node more closely by clicking on it (see Figure 11; here we clicked the blue node on the very right). Selecting reveals two things: In the graph panel, you can see the explanation scores from Section D for this node in this layer. In the tree panel, you can see the decision path in the tree for this node. This is particularly helpful if multiple leaves in the tree would lead to the same output state as in this example.

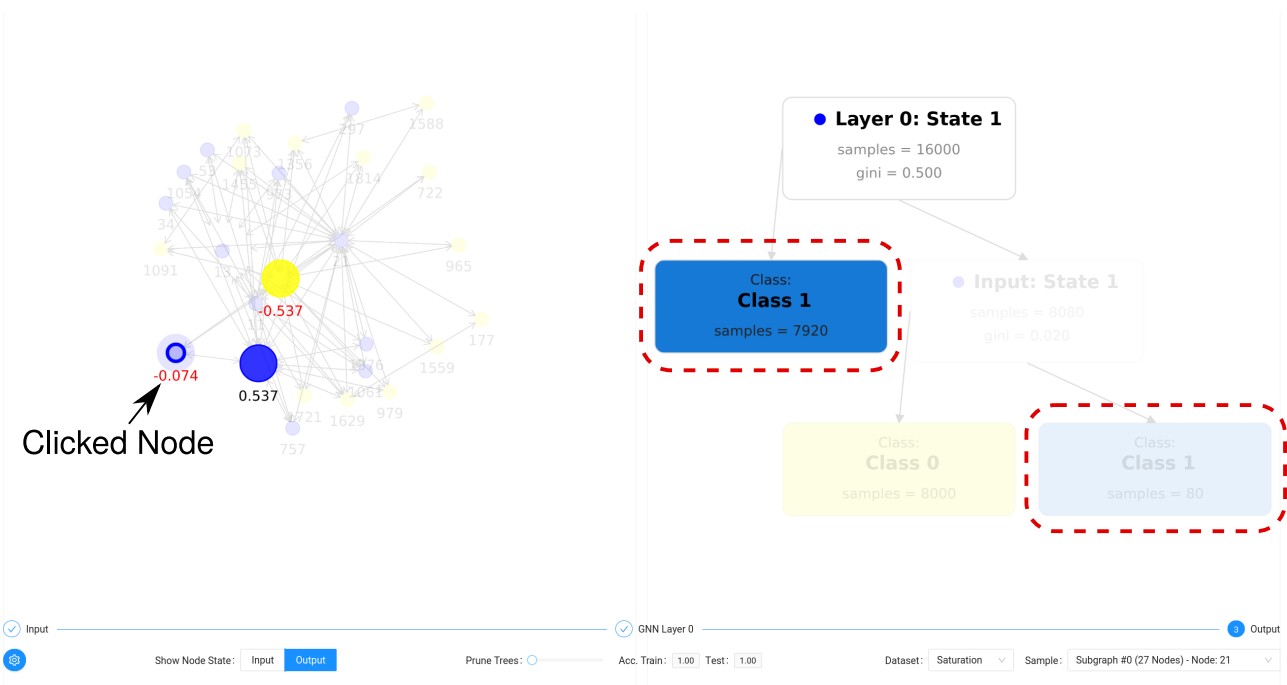

Figure 11: Interface when clicking on a node for closer examination. We can see node-level importance scores for this node on the left and the taken decision path on the right. Two paths end in the blue state, shown by the red boxes. The path the node takes is highlighted, the other path is blurred out.

# D. Generating Explanations

In this section, we describe in detail how we can use GraphChef recipes to derive importance scores for the classification of a single node/graph. As in many existing explanation methods, these scores form a heatmap over all nodes to identify important inputs.

Formally we are going to compute scores of the form $\mathbb{R}^{N \times S \times N}$ where $N$ is the number of nodes and $S$ the number of categorical states. We assume for simplicity that every layer has the same number of states. For one node $u$ and one state $s$ the explanation $e(u, s)$ is a real-valued vector that assigns every other node $v$ an importance how much $v$ contributes to $u$ being in state $s$. We accumulate the importance over layers.

Importance for every node $u$ for the encoder layer are initialized as $e(u, v) = \mathbb{1}_u$ for every state $v$, where $\mathbb{1}_u$ is a vector that is 1 at the index of $u$ and 0 everywhere else. In other words, every node is its own explanation after the encoder.

To compute the explanation update for node $u$ in a GraphChef layer, we investigate its decision tree. First, we compute the Tree-Shap values for $u$ in the decision tree. These values reveal how important each decision feature in the tree are for predicting $u$; a value of 0 corresponds to an unused decision. Depending on the type of decision feature — a state feature, a message features, or a delta feature (see possible cases in Figure 3) — we will add explanation to nodes differently. We handle each decision feature independently and weigh it with it's Tree-Shap value.

**State features.** There are $S$ possible state features that can each lead to $S$ different new states. This yields $S \times S$ Tree-Shap values that we denote with $\tau_S(s, s')$ To compute explanations we additionally require the indicator variable $\text{sign}(s)$ that is 1 if $u$ is in state $s$ at the start of the layer, and $-1$ otherwise. This indicator allows us to measure negative evidence that $u$ is *not* in a certain state. The "propagation" of state features is then easy since all importance stays with the node.

$$\sigma(u, s') = \sum_{s \in S} \tau_S(s, s') \cdot e(u, s) \cdot sign(s)$$

**Message features.** There are also $S$ message features that can lead to $S$ different states, thus we have $S \times S$ Tree-Shap values $\tau_M(s, s')$. Computing explanations for a neighbor feature gives each neighbor in the state $s$ importance, normalized by the number of neighbors. Let $N(s)$ denote $u$'s neighbors in state $s$:

$$\mu(u, s') = \sum_{s \in S} \tau_M(s, s') \cdot \sum_{v \in N(s)} \frac{e(v, s)}{|N(s)|}.$$

**Delta features.** We have $S^2 - S$ delta features where $(s, s')$ encodes the feature that there are more neighbors in state $s$ than neighbors in $s'$. Here we use the Tree-Shap values $\tau_\Delta(s, s', s'')$. We also need the indicators variable $(\mathbb{1}_{>(s,s')})$ that are 1 if indeed more neighbors are in state $s$ rather than $s'$ and 1 if not. Now, explanation for delta features is similar to that of neighborhood features, where the majority class contribution is positive and the minority class contribution is negative:

$$\delta(u, s'') = \sum_{s \in S} \sum_{s' \neq s \in S} \tau_\Delta(s, s', s'') \frac{\sum_{v \in N(s)} e(v, s) - \sum_{v \in N(s')} e(v, s')}{|N(s)| + |N(s')|} \cdot \mathbb{1}_{>(s,s')}.$$

These explanations are added to those of the previous layers:

$$e(u, s) = e(us, s) + \sigma(u, s) + \mu(u, s) + \delta(u, s)$$

**Decoder layer** The decoder layer is slightly special since it uses skip connections. For node classification, we directly concatenate all intermediate features and use the same computation scheme to compute the final explanations. For graph classification we additionally need to pool the nodes. We do this layer-wise and supply the decoder layer with per-layer node counts per state. The decoder can then use counting and comparison features similar to $M$ and $\Delta$ features in the GraphChef layers. The only difference is that instead of propagating the explanation to neighbors, we now need to propagate it to all of the nodes in the graph that were in the corresponding states.

# E. Datasets

## E.1. Synthetic Datasets

- **Infection** (Faber et al., 2021) is a synthetic node classification dataset. This dataset consists of randomly generated directed graphs, where each node can be healthy or infected. The classification task predicts the length of the shortest directed path from an infected node.

- **Negative Evidence** (Faber et al., 2021) is a synthetic node classification dataset. A random graph with ten red nodes, ten blue nodes, and 1980 white nodes is created. The task is to determine whether the white nodes have more red or blue neighbours.

- **BA Shapes** (Ying et al., 2019) is a synthetic node classification dataset. Each graph contains a Barabasi-Albert (BA) base graph and several house-like motifs attached to random nodes of the base graph. The node labels are determined by the node's position in the house motif or base graph.

- **Tree Cycle** (Ying et al., 2019) is a synthetic node classification dataset. Each graph contains an 8-level balanced binary tree and a six-node cycle motif attached to random nodes of the tree. The classification task predicts whether the nodes are part of the motif or tree.

- **Tree Grid** (Ying et al., 2019) is a synthetic node classification dataset. Each graph contains an 8-level balanced binary tree and a 3-by-3 grid motif attached to random nodes of the tree. The classification task predicts whether the nodes are part of the motif or the tree.

- **BA 2Motifs** (Luo et al., 2020) is a synthetic graph classification dataset. Barabasi-Albert graphs are used as the base graph. Half of the graphs have a house-like motif attached to a random node, and the other half have a five-node cycle. The prediction task is to classify each graph, whether it contains a house or a cycle.

| Dataset | Graphs | Classes | Avg. Nodes | Avg. Edges | Features |
|---|---|---|---|---|---|
| Infection | 1 | 7 | 1000 | 3973 | 2 |
| Negative Evidence | 1 | 2 | 2000 | 102394 | 3 |
| BA Shapes | 1 | 4 | 700 | 4110 | 0 |
| Tree Cycle | 1 | 2 | 871 | 1942 | 0 |
| Tree Grid | 1 | 2 | 1231 | 3130 | 0 |
| BA 2Motifs | 1000 | 2 | 25 | 50.96 | 0 |

Table 11: Statistics of Synthetic Datasets

## E.2. Real-World Datasets

- **MUTAG** (Debnath et al., 1991) is a molecule graph classification dataset. Each graph represents a nitroaromatic compound, and the goal is to predict its mutagenicity in Salmonella typhimurium. Mutagenicity is the ability of a compound to change the genetic material permanently, usually DNA, in an organism and therefore increase the frequency of mutations. The nodes in the graph represent atoms and are labeled by atom type. The edges represent bonds between atoms.

- **Mutagenicity** (Kazius et al., 2005) is a molecule graph classification dataset. Each graph represents the chemical compound of a drug, and the goal is to predict its mutagenicity. The nodes in the graph represent atoms and are labeled by atom type. The edges represent bonds between atoms.

- **BBBP** (Wu et al., 2017b) is a molecule graph classification dataset. Each graph represents the chemical compound of a drug, and the goal is to predict its blood-brain barrier permeability. The nodes in the graph represent atoms and are labeled by atom type. The edges represent bonds between atoms.

- **PROTEINS** (Borgwardt et al., 2005) is a protein graph classification dataset. Each graph represents a protein that is classified as an enzyme or not and enzyme. Nodes represent the amino acids, and an edge connects two nodes if they are less than 6 Angstroms apart.

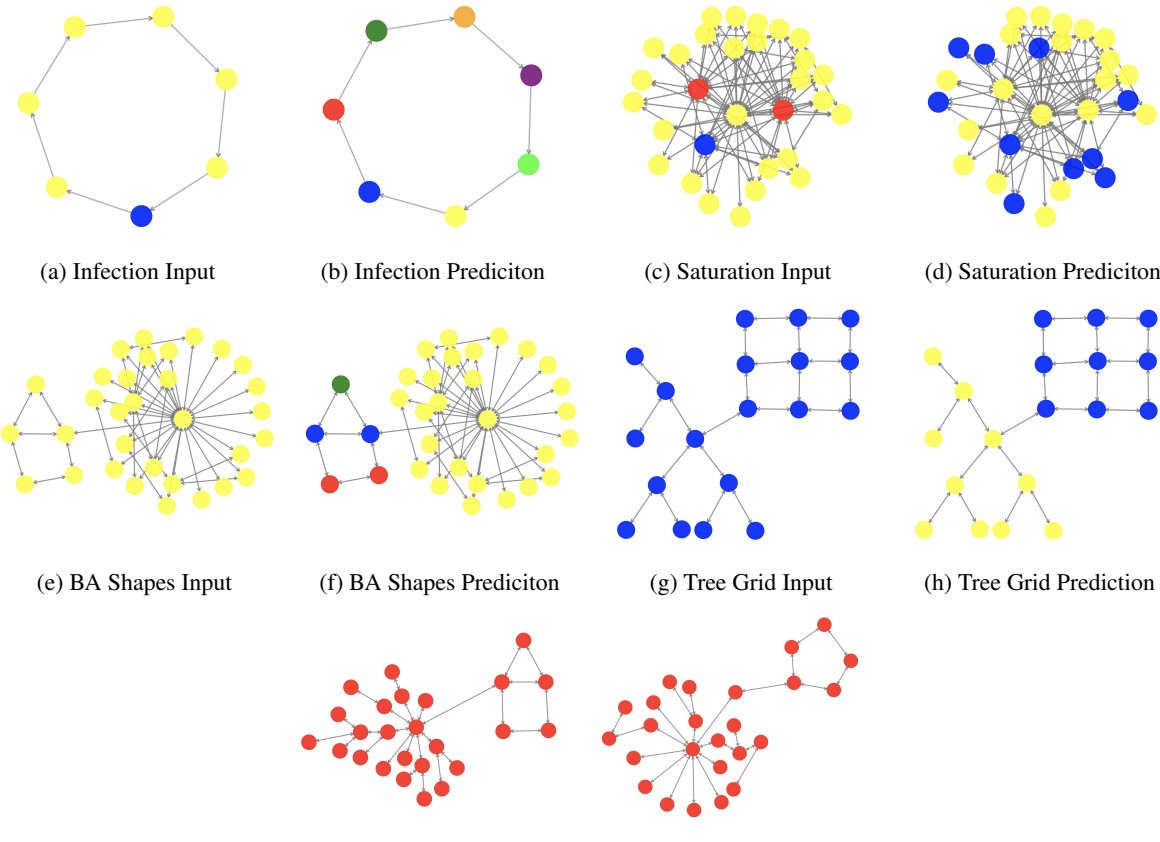

(a) Infection Input     (b) Infection Prediciton     (c) Saturation Input     (d) Saturation Prediciton

(e) BA Shapes Input     (f) BA Shapes Prediciton     (g) Tree Grid Input     (h) Tree Grid Prediction

(i) BA 2Montifs - House Input     (j) BA 2Montifs - Cycle Input

Figure 12: Synthetic Benchmarks - Example Graphs

- **REDDIT BINARY** (Borgwardt et al., 2005) is a social graph classification dataset. Each graph represents the comment thread of a post on a subreddit. Nodes in the graph represent users, and there is an edge between users if one responded to at least one of the other's comments. A graph is labeled according to whether it belongs to a question/answer-based or a discussion-based subreddit.

- **IMDB BINARY** (Borgwardt et al., 2005) is a social graph classification dataset. Each graph represents the ego network of an actor/actress. In each graph, nodes represent actors/actresses, and there is an edge between them if they appear in the same film. A graph is labeled according to whether the actor/actress belongs to the Action or Romance genre.

- **COLLAB** (Borgwardt et al., 2005) is a social graph classification dataset. A graph represents a researcher's ego network. The researcher and their collaborators are nodes, and an edge indicates collaboration between two researchers. A graph is labeled according to whether the researcher belongs to the field of high-energy physics, condensed matter physics, or astrophysics.

- **Cora**, **CiteSeer**, and **PubMed** are popular citation networks (Yang et al., 2018b). Nodes are papers and citations are edges. Nodes contain features that represent words of their contents and are labeled by sub-fields.

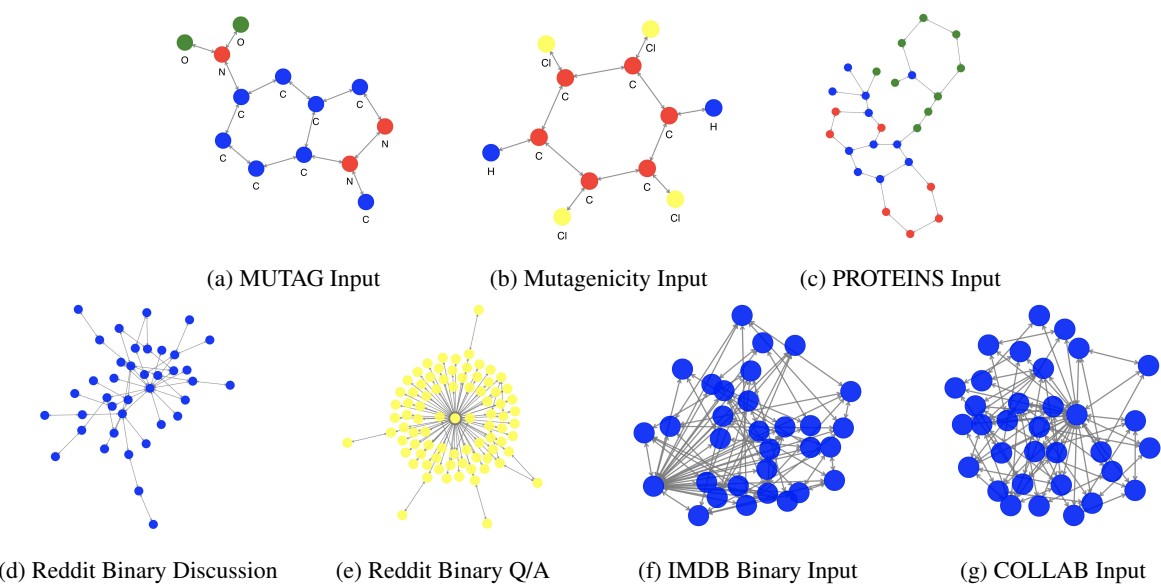

Figure 13: Real-world benchmarks - Example graphs

| Dataset | Graphs | Classes | Avg. Nodes | Avg. Edges | Features |
|---|---|---|---|---|---|
| MUTAG | 188 | 2 | 17.93 | 39.59 | 7 |
| Mutagenicity | 4337 | 2 | 30.32 | 61.54 | 14 |
| BBBP | 2039 | 2 | 24.06 | 51.91 | 9 |
| PROTEINS | 1113 | 2 | 39.06 | 145.63 | 3 |
| REDDIT BINARY | 2000 | 2 | 429.63 | 995.51 | 0 |
| IMDB BINARY | 1000 | 2 | 19.77 | 193.06 | 0 |
| COLLAB | 5000 | 3 | 74.49 | 4914.43 | 0 |
| Cora | 1 | 7 | 2485 | 5069 | 1433 |
| CiteSeer | 1 | 6 | 2110 | 2668 | 3703 |
| PubMed | 1 | 3 | 19717 | 44324 | 500 |

Table 12: Statistics of Real-World Datasets

