# OpenReview forum: "GraphChef: Learning the Recipe of Your Dataset"
_ICML.cc/2023/Workshop/IMLH — IMLH 2023 Oral_

### Official Review · Reviewer_91zT · 2023-06-10
**The authors present a novel approach for graph classification tasks, called GraphChef. This approach is based on the Stone Age model for distributed computing and has been designed to interpret the reasoning behind graph classification tasks. The authors aim to provide explanations for machine learning predictions, a crucial step in many areas that require explainable AI.**

**Rating:** 9
**Confidence:** 3

**Review:**

The work is of high quality. The approach is sound, and the methodology is scientifically robust. The authors propose a new layer for Graph Neural Networks (GNNs) based on a simplified distributed computing model and then distill this into decision trees to create GraphChef. This model is then used to abstractly express the reasoning behind a graph classification task. The authors also propose a pruning method to maintain a balance between model accuracy and simplicity. They further show how to compute node-level importance scores, which is an established method in GNN explanation.

---

### Official Review · Reviewer_LMS2 · 2023-06-11
**Review from Reviewer LMS2**

**Rating:** 5
**Confidence:** 4

**Review:**

The paper proposes a novel approach to replace the updation rule of GNNs with a decision tree. However, there are several issues that need to be clarified (see weaknesses).

**Strengths**
- The proposed decision tree base node representation updating rule is very interesting and novel.
- The proposed GraphChef framework provides direct explanations, instead of using intermediate embeddings/attentions to support explanations.

**Weaknesses**
- In Abstract, Introduction, and Proposed Approach, the authors claim that "GraphChef enables us to understand graph datasets as a whole". From my understanding, that means they would use a graph to represent the whole dataset, where each node represents an instance/sample in the dataset. However, in Figure 1, they show that each graph represents only one protein (contains 17 helix nodes and two sheets nodes). If the graph they use is as Figure 1 stated, then the GNN model does not make sense. Because "neighbors" in protein-level graphs describe how amino acid elements are connected in structure.  I strongly suggest authors to revise this part, especially distinguishing between "protein graph" and "graph" used in GNN.
- The notations in Sec 3 are confusing. From the original GIN, the aggregation rule first gets $a_v^{l+1}$ from the node itself and its neighbors and then gets the next layer node representation $h_v^{l+1}$ from $a_v^{l+1}$. However, authors use $h_v^{l+1}$ both in the aggregation rule (L163 in the right page column) and updating rule (L207 in the left page column), which makes the technical part hard to follow.
- In Table 1, both GIN and dish-GIN achieve perfect performance (test-acc=1.0) in the first six datasets. I am a bit concerned about it. Are these datasets too easy?

---

### Official Review · Reviewer_4FBE · 2023-06-11
**An tree based explanation for graphs**

**Rating:** 7
**Confidence:** 3

**Review:**

This paper proposes a new method for explanation of GNNs using trees. The method is sound and novel. However, I think the way the authors pose their results is not accurate. They claim their method (presumably unlike other explanation method) is looking into entire dataset for reasoning: "Existing GNNs and explanation methods reason on individual graphs not on the entire dataset". However, the motivation example is not clear: "So far GNNs are used as black-box models to classify individual graphs, as in “Is this protein (represented as a graph) an enzyme?” In contrast, in our work we do not only want to understand individual graphs, but whole datasets, as in “What makes a protein an enzyme?"." If an explanation method used for a single graph (which is an enzyme) explains why the model thinks this is an enzyme, this would explains "what makes a protein an enzyme". And clearly, such an explanation method is trained on "entire dataset". Hence, at least from this example, it is not clear why their approach is different. Overall, the paper presents an interesting method.

---

### Meta-Review · Program_Chairs · 2023-06-19

**Recommendation:** Accept (Oral)
**Confidence:** 4

**Metareview:**

This paper proposes a novel interpretation method for GNN using trees. The overall quality is good. The authors are encouraged to incorporate reviewers' feedback to further improve the writing.

---

### Decision · Program_Chairs · 2023-06-20

Accept (Oral)